# A Versatile Board for Event-Driven Data Acquisition

**DOI:** 10.3390/s24051631

**Published:** 2024-03-01

**Authors:** Gabriele Manduchi, Andrea Rigoni, Luca Trevisan, Tommaso Patton

**Affiliations:** Consorzio RFX, Corso Stati Uniti, 4, 35127 Padova, Italy; andrea.rigoni@igi.cnr.it (A.R.); luca.trevisan@igi.cnr.it (L.T.); tommaso.patton@igi.cnr.it (T.P.)

**Keywords:** data acquisition, FPGA, event-driven acquisition

## Abstract

Event-driven data acquisition is used to capture information from fast transient phenomena typically requiring a high sampling speed. This is an important requirement in the ITER Neutral Beam Test Facility for the development of one of the heating systems of the ITER nuclear fusion experiment. The Red Pitaya board has been chosen for this project because of its versatility and low cost. Versatility is provided by the hosted Zynq System on Chip (SoC), which allows full configuration of the module architecture and the OpenSource architecture of Red Pitaya. Price is an important factor, because the boards are installed in a hostile environment where devices can be damaged by EMI and radiation. A flexible solution for event-driven data acquisition has been developed in the Zynq SoC and interfaced to the Linux-based embedded ARM processor. It has been successfully adopted in a variety of data acquisition applications in the test facility.

## 1. Introduction

The ITER Neutral Beam Test Facility (NBTF) [1] aims at developing one of the two heating systems of ITER [2], the largest nuclear fusion experiment in the world, currently under construction in France. In this heating system, a beam of negative H^−^ ions is accelerated at an energy of 1 MeV. The negative ions beam is then neutralized in order to be directed to the high-temperature gas, called Plasma, where the nuclear fusion reactions occur. This neutralization process prevents distortion of the beam due the high electromagnetic fields required to confine the plasma inside its container. Many technological challenges (several of them not yet resolved) must be faced in the development of a neutral beam injector using an acceleration voltage of 1 MV and with a final beam power of 16 MW. A large test facility has been therefore built in order to develop and test possible technological solutions requiring a sophisticated control and data acquisition system (CODAS) [3]. It must be able to collect a variety of measurements in order to support the project and the development of its critical components, such as the 1MV voltage generation, the H^−^ ion source, the proper beam focusing and its neutralization and the dissipation of the generated power. 

Within CODAS, various measures are undertaken, and addressing fast transient phenomena necessitates ad hoc solutions to enable event-driven high-speed data acquisition. In order to capture the required information for fast transient phenomena, such as the breakdowns that from time to time occur in the acceleration grids, a very high sampling frequency is required. It would, however, be unfeasible to store data from continuous ADC sampling at this speed, as the beam can be generated for a time lasting up to one hour. For this reason, a measuring system able to detect the occurrence of such phenomena (called events) and to trigger fast data acquisition for a given region of interest (ROI) centered around the event occurrence is required. As the occurrence of events is unpredictable, and due to the potential cascading of events, the data acquisition system must be able to be quickly retriggered and must provide the ability to withstand sustained data flow when cascades of events occur. The required sampling rate for such transient phenomena is in the order of 100 MHz in order to capture enough information for subsequent analysis. 

Many Analog-to-Digital Converter (ADC) devices are available on the market that allow for acquiring data and storing them in the computer memory. Very few commercial devices, however, provide streamed event-driven data acquisition, and, indeed, one of these devices was initially adopted [4] in the NBTF test facility. However, due to the lack of a flexible triggering mechanism and an ROI size limited by available memory size, a new project was initiated to develop a more adaptable module for event-driven data acquisition. In particular, we addressed the following aspects:A more flexible triggering mechanism incorporating not only external trigger sources, but also internal triggers generated by given conditions on the signal level, in order to automatically start an event readout, possibly triggering other devices.An ROI size not constrained to the internal memory size, but one that is able to sustain post-trigger samples streaming. In this scenario, the number of pre-trigger samples will still be limited by the size of the circular memory buffer, but the number of post-trigger samples is no longer limited by such size, provided the data transfer rate can withstand the sampling frequency.A flexible usage of internal and external clocks. External clocks are required to ensure timing consistency among waveforms acquired by different devices, because of the unavoidable clock drift when using internal clock sources. However, the timing system currently used in the test facility limits the frequency of the distributed clock to 1 MHz, and data acquisition at higher frequencies would require PLL in order to be locked in phase. When handling event-driven data acquisition, in our case, it is important to discriminate events and to discard events that are triggered by spurious noise rather than by real physical events. Events recorded by different devices within a time window of 1 µs can be safely assumed to be originated by a real event, taking into account the dimension of the experimental plant (~100 m) and the speed of light. Timestamping event occurrence based on the 1MHz external clock provides, therefore, enough precision without implementing additional PLLs. Moreover, internal clocks can be used for fast data acquisition during the ROI interval, being the clock drift among devices negligible within the ROI time window.

In order to fulfill the above requirements, we searched for an Analog-to-Digital Converter (ADC) equipped with a programmable FPGA. The Red Pitaya (RP) [5] board has been considered for several reasons, among which:*Availability of high-speed ADC converter*. The 125 MHz sampling speed that can be achieved for two ADC channels available in the RP board is adequate for capturing the relevant information, such as rise time, overshoot and oscillations.*Zynq architecture*. The Zynq SoC used in the RP board is well suited for our applications, where fast signal handling, such as triggering and circular buffer management, is carried out in FPGA, and other functions, such as overall management and data transfer, are carried out by the embedded ARM processor. The available RAM (2 or 4 GB) allows temporary data storage, and the 1Gbit Ethernet connection allows full remote control.*Reduced cost*. Cheap solutions are always preferred over expensive ones, but in our case, this fact is further stressed as the RP boards are hosted in a hostile environment with high risk of electrical damage, such as the 1 MV high-voltage deck. As a consequence, a mortality rate due to electrical discharges has to be taken into account.*Reduced board form factor and insulation capability.* The overall design of the RP board is meant to replicate the success of small-factor devices, where solutions that are easy to carry and plug represent a key to success. This proved to also be effective within our experimental environment, where a spread location deployment is needed to acquire data in a large facility, and where the high-voltage application very often asks for a strong electrical insulation. In these situations, we indeed used the complete RP as an isolated device, linked with the network via fiber media converters and powered with separated isolation transformers or even with battery supply.

In the rest of the paper, the motivations for the adopted approach are presented in Section 2. The SoC architecture, i.e., the FPGA and its integration in the Linux-based embedded processor, are presented in Section 3. Section 4 presents some performance measurements. Section 5 discusses how the RP boards have been integrated in the overall data acquisition. Section 6 presents an overview of the application of the developed board in the test facility, and Section 7 draws the final considerations, also presenting other usages (present and foreseen) of the implemented architecture. 

## 2. Motivations

The RP board represents a cheap and flexible alternative to other expensive solutions for a variety of instrumentation tools. The board hosts two 125 Msps 14-bit ADC inputs and two 14-bit Digital-to-Analog Converters (DAC), a XILINX Zynq 7010 SoC, including a 2-core ARM processor, USB and 1Gb Ethernet connectors. In our application, Linux is installed in the processor in order to balance the workload between the processor and the FPGA, letting time-critical functions be carried out by the latter and the rest of the computation and network communication be carried out by the former.

The versatility of the Zynq architecture is paid by the complexity in the development, requiring skills in Hardware Description Language (HDL) and Linux driver development. This difficulty is overcome by development frameworks that allow developing a measurement system without any knowledge of FPGA and Linux driver programming. Two such frameworks have a widespread usage: SCPI and PyRPL. SCPI (Standard Commands for Programmable Instrumentation) [6] is a framework that allows the RP board to be controlled remotely over a LAN or wireless interface using MATLAB, Scilab or Python. The SCPI commands are recognized by the instrument and the corresponding action taken, such as acquiring data from the fast input ADC, generating signals or controlling other peripheral devices. This approach allows for quickly writing control programs in MATLAB or Python, and it has been adopted, for example, in [7] for measurement in optical wireless communication and in [8] for electrical impedance tomography systems. PyRPL [9] is an open-source software package providing several instruments on the RP board, such as oscilloscopes, network analyzers, lock-in amplifiers, feedback controllers and digital filters. PyRPL has a python interface and an associated Graphical User Interface (GUI). Instruments are instantiated as python classes that can be configured either programmatically or via the associated GUI. PyRPL has been used, for example, in [10] for controlling quantum optic experiments and in [11] for digital lase frequency and intensity stabilization. Both SCPI and PyRPL aim to minimize effort and development time, obtaining results that could previously be obtained only using expensive instrumentation. As such, the two frameworks are perfect for applications where different components must be composed to fulfill laboratory tasks. The flexibility and ease of use are counterbalanced by some drawbacks that prevent their usage in our event-driven streamed data acquisition, namely: *Limits in the available functions*, confined to the available components in the framework. Despite the richness of the component toolbox, no available component could fit our specific requirements.*Limits in performance*. Performance is a key factor in our project because the event-driven acquisition must be able to face cascades of events that may occur at a high rate. While the frameworks allow composing components, the way data flow is managed is transparent to the user and cannot be optimized.

For this reason, we directly implemented both the FPGA program and the Linux driver in order to provide all the necessary functionality and to optimize data transfer performance. Direct FPGA implementation in RP applications has been carried out, for example, in [12] for frequency stabilization and spectroscopy parameter optimization and in [13] for the demodulation of high-frequency atomic face microscopy probes. 

## 3. Architecture

The following parameters define the behavior of the event-driven data acquisition device:*Pre- and post-trigger samples*: samples before and after the trigger to be acquired. While the number of pre-trigger samples is limited by the dimension of the used circular buffer (block memory), the maximum number of post-trigger samples is, in principle, unlimited, provided the data flow rate in the readout is fast enough to avoid the circular buffer overflow (see below).*Trigger mode*: specifies the trigger logic. It can be an external digital signal or internally detected, based on some conditions on the level of the input signal, in this case; however, an external trigger must be provided in order to enable the triggering logic and to properly timestamp trigger occurrences. The trigger can be single, i.e., the device is triggered once after being armed, or multiple, i.e., the device is allowed to be quickly retriggered after the ROI has been acquired.*Clock mode*: specifies how acquired samples are timestamped. Its possible values are
○*Internal*: sampling and trigger timestamping are derived from the internal 125MHz FPGA clock. The lower sampling speed is specified via a clock divide register.○*External*: sampling and trigger timestamping are derived from an external clock signal.○*External trigger*: sampling is driven by the (possibly divided) internal clock, while trigger timestamping is derived from an external clock signal. This feature is useful in our application for long-lasting event acquisition, where sporadic events must be recorded over a long period (up to one hour). The acquisition of the single events requires high sampling frequency, much larger than the 1 MHz external clock reference used for inter-module synchronization. In this case, event timestamping is based on the external clock, while the high-frequency sampling clock is internally generated.

The device can be programmed via software by means of a set of registers, i.e., memory addresses that are mapped against internal FPGA locations. Some registers (named *Mode* register, *PreSamples* register, *PostSamples* register, *ClockDivision* register, respectively) define the mode of operation, while a *Command* register will be used to arm, software trigger and stop the device. As commonly adopted in transient recorders, a circular buffer (implemented in FPGA by an available XILINX component named Block Memory IP [14]) is used to store the recent history of the input signal. Two pointers, *PrePtr* and *PostPtr* are used: *PostPtr* is incremented to the next position by the FPGA logic every time an input sample is received, after storing the sample in the buffer in the location defined by that pointer. When the trigger logic detects a new event, *PrePtr* is set by the FPGA logic to the current value of *PostPtr* minus the number the content of the *Pre* register (in a circular way), so that *PrePtr* and *PostPtr* at this instant locate the set of pre-trigger samples in the circular buffer. At the trigger time, another asynchronous process in the FPGA is started. This process monitors the two pointers at every FPGA clock cycle, and if the two pointers are different, it copies the buffer value corresponding to the address specified by *PrePtr* to the output FIFO and advances that pointer (see Figure 1). After a number of samples corresponding to *PreSamples* plus *PostSamples* have been transferred to the output FIFO, the process suspends until activated by a new trigger. In this way, provided the output FIFO does not overflow, the number of post-trigger samples is not limited by the length of the circular buffer. Therefore, it is possible to also carry out continuous (streamed) data acquisition, with the only limit being that the sampling speed rate cannot exceed the achievable data transfer rate. 

Readout data are copied to the output FIFO in order to withstand data bursts after event detection. Data can be transferred to the processor in two ways:(a)Directly to the Linux driver via a FIFO component (XILINX AXI Stream FIFO IP [15]) mapped onto memory registers. An interrupt is generated when data are available, and then the FIFO is directly read by the driver code.(b)Via Direct Memory Access (DMA). In this case, the DMA controller will issue an interrupt when the ROI has been acquired.

The XILINX Vivado Integrated Development Environment (IDE) [16] and the ANACLETO framework [17] for orchestrating the toolchain in the project are used to generate the FPGA configuration. The FPGA definition is composed of several modules written in the VHDL Hardware Description Language and connected with other available Intellectual Properties (IP) cores. The overall block design is then compiled in order to produce the FPGA bitstream and the hardware configuration expressed in a Linux Device Tree. The bitstream and the device tree are then copied into the SD card mounted on the RP board in order to carry out the specified functions. In addition, it is necessary to write the code of the Linux driver exporting the FPGA functionality to user level programs. Writing Linux device drivers is not trivial, and every change in the FPGA configuration affecting any interface component, such as a register, FIFO or DMA controller, requires a corresponding change in the device driver code. In order to speed the development process, we developed a driver code generator based on the textual form of the generated device tree available in a file generated by VIVADO during the generation of the bitstream. Linux driver code generation is based on a generic template driver code that is then expanded by a python program that parses the device tree description and expands placeholders in the code template. The generated Linux driver defines and exports a set of *ioctl* calls for every declared register, FIFO interface and DMA controller defined in the VIVADO project. Recall that *ioctl()* is one of the standard routines that programmers use to interact with devices. The other main routines are *open(), read()* and *write()*. In particular, *ioctl()* is used for all the driver operations different from read and write, and it accepts as argument a command that indicates the requested operation. Table 1 lists the generated *ioctl* commands for every different component. Moreover, if data readout is implemented via direct FIFO readout, the generated driver code will implement the *read()* call, synchronizing it to the availability of data samples.

## 4. Data Transfer Performance

Thanks to the internal circular buffer management, acquired data bursts can be larger than the size of the circular buffer itself, provided that data can be read at a speed that is enough to avoid the overflow of the readout FIFO. It is therefore possible to also implement continuous (streamed) data acquisition, and, indeed, this is supported by providing a special value to the Post-Trigger Samples register. The possible reasons for readout FIFO overflow are of two types:1.Burst overflow, occurring when data coming from a single burst (whose size is PreTriggerSamples + PostTriggerSamples) are transferred to the readout FIFO in a time that is too short in respect to that required for reading data from the FIFO. This occurs when a very high sampling frequency is selected and the sum of pre and post samples exceeds the size of the readout FIFO that has been set to 16 kBytes due to the limitation of available FPGA memory resources in the Zynq 7010 System used in the RP board.2.Streaming overflow that occurs when the average data throughput, i.e., the event occurrence frequency multiplied by the burst size, is larger than the average data transfer from the readout FIFO.

Two different data transfer techniques have been implemented. The first one is based on DMA transfer and minimizes the number of CPU cycles required to carry out data transfer. Double buffering is also provided in order to decouple the copy from the DMA buffer from the concurrent device-to-memory DMA transfer. This technique maximizes data throughput, allowing a maximum sustained data transfer throughput of 40 MSamples/s. 

While DMA transfer is optimal for data acquisition, it is not suitable in cases where ADC samples are used in real-time control, i.e., when using the RP board to close a control loop using the available high-speed DAC channels. The control loop can be closed in two ways:1.Carrying out control computation directly in FPGA. This solution allows for the fastest control but is limited in computation. Simple PID control can be implemented in this way (and indeed PID blocks are available in the PyRPL toolkit), but more complex algorithms requiring floating point computation cannot be implemented directly in FPGA.2.Carrying out control computation in the embedded ARM processor. In this case, it is necessary to transfer data from the FPGA into processor memory and vice versa.

When using the second option, DMA-based data transfer is normally not feasible because the whole DMA transfer must be terminated before its data can be used for control computation. This introduces a delay in the data availability that may not be tolerable in control. For this reason, direct FIFO readout may be required, i.e., reading data samples from the FIFO as soon as they are available. This solution reduces latency and jitter in data readout but introduces a penalty in data transfer throughput because several CPU cycles are required for reading every data sample. Figure 2 provides an indication of the data throughput that can be achieved with direct FIFO readout. In the figure, the maximum burst length that can be achieved without FIFO overflow is shown at different sampling rates. For the highest sampling rates (up to 125 MHz), the maximum burst length corresponds to the FIFO size because, in this case, the data acquisition speed is much higher than the speed at which the FIFO is read by the processor. When the sampling rate decreases, the burst size can be larger than the FIFO size because at the same time as data are acquired and written to the FIFO, they are concurrently read by the processor. For sampling rates less than 1.2 MHz, the average data readout exceeds the data sampling rate, and sustained data transfer can be achieved. It is worth noting that only in sustained data transfer, i.e., at a sampling frequency less than 1.2 MHz, is feedback control guaranteed not to suffer delays in data availability due to the internal buffering. In this case, as soon as a sample has been stored in the internal circular buffer, it will be transferred to the processor before a new sample is acquired. 

## 5. Data Integration

The flexibility of the Zynq architecture of the RP boards allows the easy integration of acquired data into a more general data acquisition system. Once transferred in the processor memory, either in DMA or via direct FIFO readout, data from the different RP boards can be transferred over the network to a central data acquisition system and thus become accessible for analysis and display. This approach has been followed in the integration of the RP boards in the NBTF test facility and pushed further by storing in the experiment database not only the acquired signals, but also the configuration of the RP boards. This has been made possible by allowing every RP board to be remotely configured via TCP/IP. Interfacing all the involved RP boards in the data acquisition and management system used in the experiment greatly simplified their usage because it was possible to rely on the user interfaces provided by the central data acquisition system for board set-up and to use the visualization tools to display the acquired signals. An example of the interface is shown in Figure 3, where a specific form is activated when selecting the corresponding item in the graphical configuration browser of the experimental set-up.

## 6. Applications

Data acquisition based on the RP board has been used in three main applications within the NBTF experiment:

*Breakdown measurement*, aiming at detecting the currents flowing during breakdowns, which are discharged from time to time along the beam acceleration grids. For this purpose, a set of RP boards has been used to measure the breakdown current in different locations inside the vacuum vessel of the NBTF test facility, both to identify the regions most involved in the discharge and to validate the fast transient model of the High Voltage plant. In this application, the RP have been integrated in a distributed trigger network, which distributes the trigger to all the interconnected devices and synchronizes the measurements. 

1.*Spectral analysis of the Radio Frequency sources*. In this measurement, the RP board has been used to provide, in real time, spectral analysis of signals from the Radio Frequency (RF) source used to ionize hydrogen in the beam source. In this application, FFT analysis is performed on the fly during data acquisition in order to acquire spectral information of the acquired RF signal. This is achieved by acquiring at high speed (125 MHz) bursts of data samples that are triggered at a constant rate of 1 kHz. The acquired bursts are then transferred in real time to the ARM processor, where FFT analysis is carried out on the fly for that bunch of data and then transferred to the central CODAS via the network.2.*Beamlet current measurement*. The H^−^ ion current beam accelerated by the acceleration grids has been divided in a number of beamlets that cross the grids. Figure 4 shows the holes in the acceleration grids for letting the beamlets traverse it.

Beamlet currents in different positions have been measured using two different configurations of the RP board. Continuous data acquisition provides information about the average current density and uniformity [18], whereas the event-driven acquisition provides information about the beam current fluctuations (MHz range) due to the plasma oscillations produced by the RF field generating the plasma [19,20]. Two different sensors have been used for continuous and event-driven data acquisition, respectively, as shown in Figure 5. Closed Loop Direct Current Current Transformers (DCCT) (1) have been used to measure the DC/low frequency components of the beamlet current, while current transformers (2) have been used to measure the high-frequency components [21,22].

A simplified scheme of the NBTF experiment showing both the main parameters affecting the measured current and the location of the beamlet current sensors is reported in Figure 6. The negative ions are produced inside an Ion Source by ionization induced by Radio Frequency (RF) field (via RF coils), then the ions are extracted from the ionized gas (Plasma) and accelerated by an electrostatic accelerator composed of three grids (Plasma Grid, Extraction Grid and Grounded Grid). The ions are extracted from the Ion Source (biased by the AGPS power supply with respect to ground) by the Extraction Grid, which is biased with respect to the Plasma Grid by a dedicated power supply (ISEG) and accelerated up to the last grid (Grounded Grid) of the accelerator. The transverse magnetic filter field, controlling the negative ion density in the proximity of the Plasma Grid, is also shown. The sensors measuring the beamlet currents located downstream the Grounded Grid in some specific locations are also shown.

In Figure 7, an example of the application of the continuous acquisition mode, implemented by the RP boards, is reported. The capability of this system to correlate the measurements collected from different sensors with other Experiment Parameters is particularly useful for data analysis, providing a powerful tool for the sensors’ integration into the experiment. The figure clearly shows how the RP board, able to record and store and synchronize the measurements in the Experiment database, is a powerful tool to clearly identify how the pulse parameter affects the measured quantity from a given sensor (a DCCT sensor, in this case). In the figure (top), the output voltage of the sensor is shown and compared with the main parameters of the pulse (Acceleration Voltage, Extraction Voltage, Magnetic filter field and Radio Frequency power).

Other applications concerning the FAST acquisition mode are shown in detail in [20].

Finally, the internal assembly of the data acquisition module is shown in Figure 8, where the layout of the six RP boards also assures high insulation capability (5 kV for this case) among the channels.

## 7. Conclusions

The flexibility of the RP board turned out to be very effective for the development of a high-speed event-driven ADC device to be used in the ITER Neutral Beam Test Facility. The RP device has been successfully used not only to measure signals derived from breakdown events in the high-voltage components, but also for several other data acquisition applications, ranging from fast bursts to slow, continuous data acquisition. The availability of a dual core embedded processor in the Zynq architecture has allowed complex operations to be carried out by a full-fledged Linux environment while hosting simpler, time-critical functions in the FPGA. Moreover, the availability of Wi-Fi through the dongle has allowed the use of the RP boards in segregated beam components, such as the high-voltage decks for which a wire connection was not feasible. It is worth noting that Wi-Fi communication has been used for data readout, not to exchange breakdown triggers among RP boards. In the latter case, direct fiber optic communication has been used in order to ensure fast and deterministic trigger times.

We have also encountered a few drawbacks in the current RP board, such as the fact that the board cannot be started without an internal SD and that a version of the board with the industrial-grade Zynq 7K (temperature and EMC) for critical applications is not yet available in the market. Moreover, only 3V3 ports can be used for IO due to the single power supply of the FPGA banks, and it has not been possible to integrate IEEE1588 time synchronization in the RP due to a limitation in the architecture of Zynq 7010/7020.

Developed in order to speed the development of Zynq-based architectures, the automated generation of the Linux driver code based on the Hardware description (device tree) generated from the FPGA configuration proved very effective, and it has also been used in other projects based on Zynq, such as a new intelligent board for the acquisition of electromagnetic probes in nuclear fusion experiments.

Finally, a summary of the RP applications in the NBTF test facility is available on the Red Pitaya website [23].

## Figures and Tables

**Figure 1 sensors-24-01631-f001:**
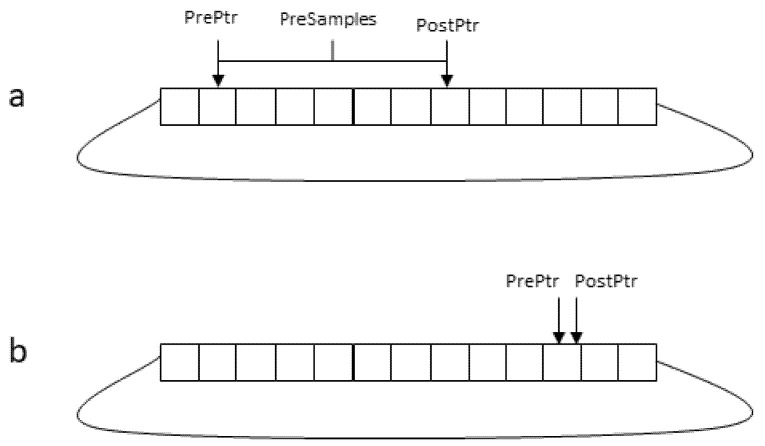
(**a**) Just after an event occurrence, the distance between *PostPtr* and *PrePtr* is equal to the number of Pre-Trigger samples. Afterwards, samples are transferred to the output FIFO. *PostPtr* is advanced every time a new sample is available, and *PrePtr* is advanced at every FPGA clock cycle, so it eventually reaches *PostPtr* (**b**) and then advances in step with it. Only when the sampling clock is equal to the FPGA clock (125 MHz) does the distance between the two pointers remain unchanged.

**Figure 2 sensors-24-01631-f002:**
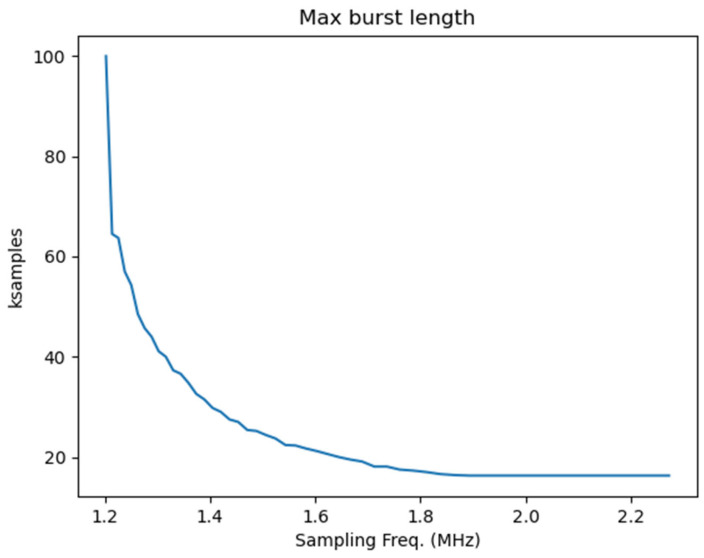
Maximum burst size when directly reading data from FIFO. For higher frequencies (up to 125 MHz), the maximum burst size is limited by the FIFO size (16k Samples).

**Figure 3 sensors-24-01631-f003:**
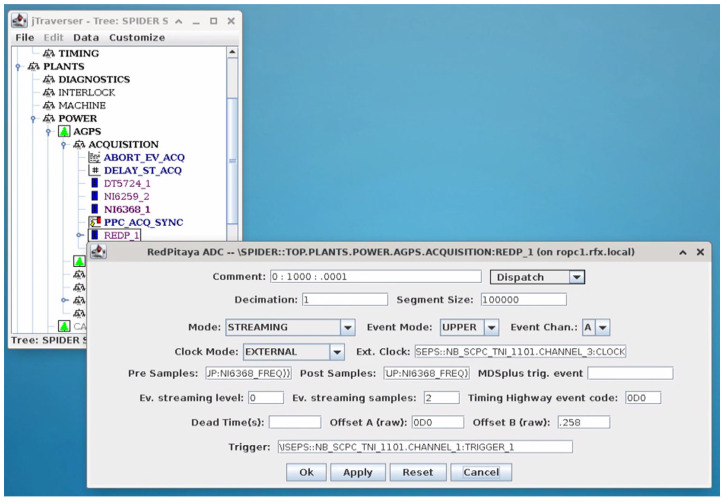
The graphical user interface for the RP board configuration.

**Figure 4 sensors-24-01631-f004:**
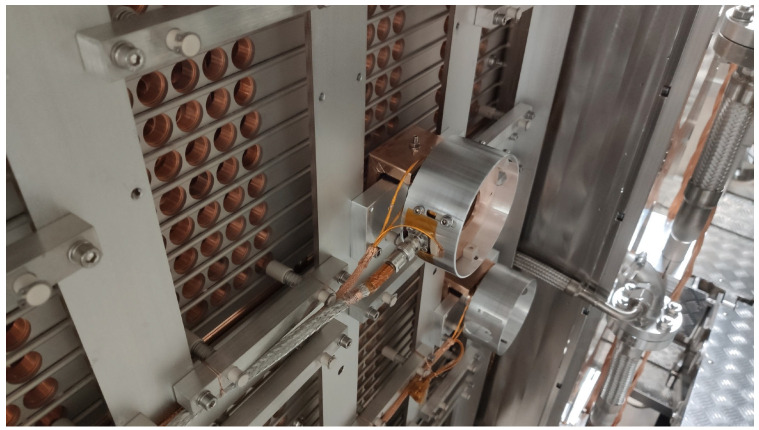
The holes in one accelerating grid that are crossed by the beamlets. The two cylinders visible in the figure host the two sensors used to measure the ion current of two selected beamlets.

**Figure 5 sensors-24-01631-f005:**
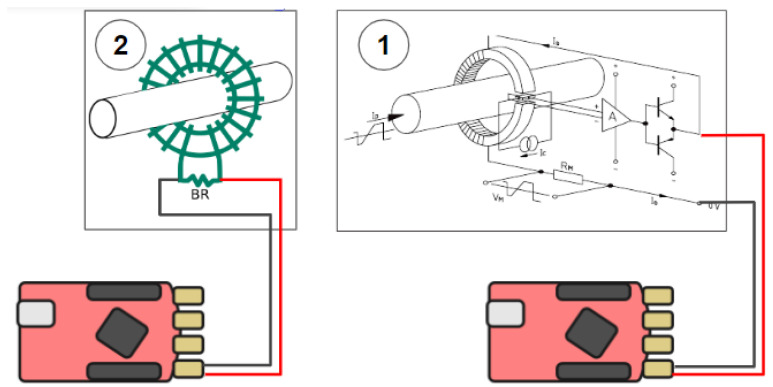
The two different measurements of the beamlet current.

**Figure 6 sensors-24-01631-f006:**
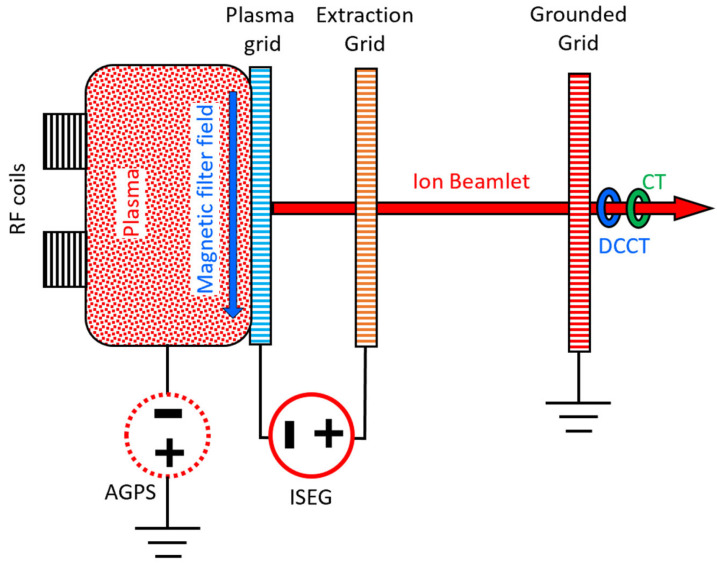
Simplified scheme SPIDER experiment where the main systems affecting the output of the sensors of the beamlet current measurement system are shown.

**Figure 7 sensors-24-01631-f007:**
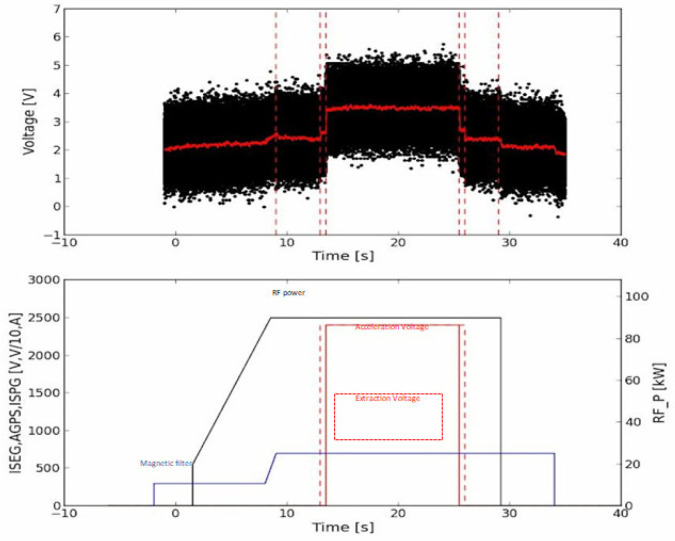
Application of continuous acquisition mode acquisition for measuring the beamlet current with a DCCT sensor. The acquired signal from the RP board connected to the sensor is shown above. The parameters affecting the sensor output (RF power, magnetic filter, acceleration and extraction voltages) are shown below, correlated by time.

**Figure 8 sensors-24-01631-f008:**
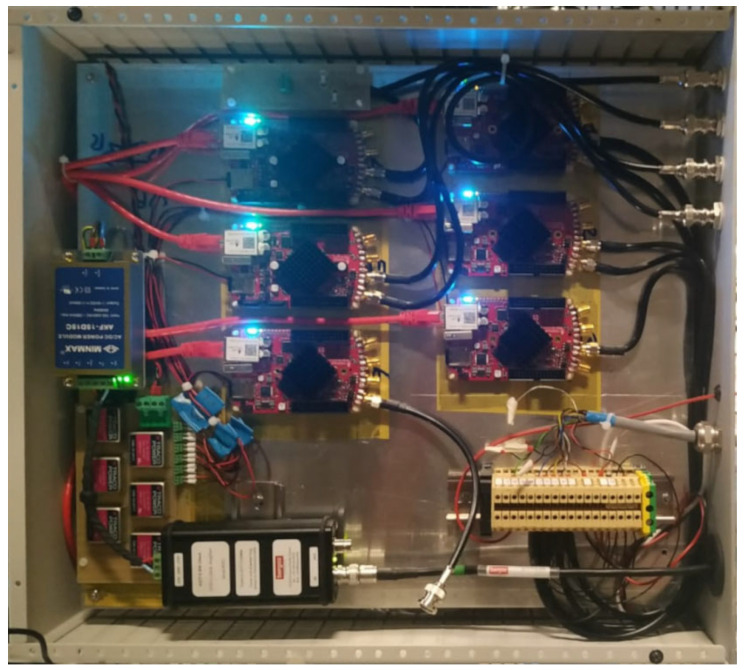
Internal assembly of the acquisition module with the Red Pitaya boards.

**Table 1 sensors-24-01631-t001:** IOCTL commands in the generated Linux driver based on the textual description of the device tree.

Component	ioctl Commands	Comment
Register	<device>_GET_<register> <device>_SET_<register>	Configuration registers declared in the FPGA configuration are mapped against pairs of *ioctl()* commands for reading and writing the 32 bit register value. <device> is the name assigned to the whole device. <register> is the name assigned to the register in the FPGA configuration
DMA Controller	<device>_SET_DMA_BUFLEN <device>_GET_DMA_BUFLEN <device>_ARM_DMA <device>_START_DMA <device>_STOP_DMA <device>_GET_DMA_DATA	If a DMA controller is declared in the FPGA configuration, the corresponding set of *ioctl* commands is generated
FIFO interface	<device>_CLEAR_<fifo> <device>_GET_LEN_<fifo> <device>_GET_VAL_<fifo>	For every declared FIFO in the FPGA configuration, the corresponding set of *ioctl* commands is generated. If one FIFO instance is used for synchronizing data readout (i.e., not using DMA), the generated read() implementation will synchronize on data availability for that FIFO. <fifo> is the name assigned to the FIFO component in the FPGA configuration.

## Data Availability

Data are contained within the article.

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
