# Peer review of "A Versatile Board for Event-Driven Data Acquisition"

_sensors, 2024, doi:10.3390/s24051631_

Round 1
Reviewer 1 Report
Comments and Suggestions for Authors
The paper presents a very interesting solution for event-driven data acquisition for ITER NBTF.
Red Pitaya provides a lot of versatility at a very low cost, which makes it a very interesting system in this type of environment where the probability of equipment damage is high.
Although it is not possible to provide all the required functionalities with a Red Pitaya, such as accurate timestamping, it can be considered an interesting solution in the state of evolution of current systems.
Congratulations for the paper, I consider it interesting for other groups and I recommend its publication.
Author Response
The authors are grateful to the reviewer for his/her comments. Even if no changes have been requested by the reviewer, the paper has been improved by fixing some typos and answering to the concerns raised by the other reviewers.
Reviewer 2 Report
Comments and Suggestions for Authors
Manuscript: 2668287
Title
A versatile board for event driven data acquisition
General comments
My feeling is that the development is mainly focused on software, which exploits the powerful features of the Red Pitaya board. Indeed the authors have done a relevant job, but I have a few perplexities.
- Apparently there is no development of sensors, just a complex software development that by the way looks prominent and makes use of other existing sensor(s) and the 125 MHz sampling ADC onboard of Red Pitaya.
- The description of the work done focuses on the developed software (including FPGA programming), with some interesting features like the automatic driver generation.
- However, such a description is too specialistic and makes use of too many acronyms and jargon, not always explained.
- The manuscript in the current form sounds more like an internal status report for the experts who developed it, rather than an article for a scientific journal.
- Here and there I found some typos and weird wording, but this is just a minor concern.
I have two suggestions.
1) Rewrite the manuscript in a form more understandable to a wider audience.
and/or
2) possibly submit it to a more specialized journal, I am afraid that Sensors could not be the right choice.
Anyway, I do not question the validity of the job done, that surely sounds of high level.
My recommendation would be rejection, but I indicate Major Revision.
I list below a few specific comments.
line 45
...sampling rate FOR such transient phenomena...
lines 173-189
This paragraph sounds unclear, I think it needs to be rewritten in a more schematic fashion, better specifying the roles and locations of Pre and Post pointers, of the samples, who and how moves them. 
e.g. Two pointers, PrePtr and PostPtr are used: the second one is moved to the next position... Does "is moved" mean physically copied elsewhere or simply that its value is incremented in order to point elsewhere?
Moreover, it is not clear whether the samples are on the circular buffer or elsewhere and the pointers are used to locate them.
lines 204-223
Too many acronyms, the general reader will be immediately lost. In order to reach a larger number of readers the content should be more understandable. The way it is now it is basically reserved to the real experts of low-level software developers. 
An effort to simplify the text, or in alternative to improve the description, would be beneficial.
Table 1
Same as above for the table.
Lines 233-238
Again, this sounds like low-level programmers jargon. For instance I doubt that the general reader will understand the statement "...due to limitation of available LUTs components in the Zynq 7010 SoC used in the RP board."
I do not know the Red Pitaya system and even though I have a solid experience in low-level programming I struggle to follow the text that seems directed to experts of Red Pitaya.
Lines 306-311 and fig.4
Closed Loop Hall Effect transducers?
rogowsky coils?
What are these transducers? Any reference?
Moreover, the figure as it is is just for the show: very small, no details, totally incomprehensible.
Lines 327-332
Sounds like nerd jargon, not for general readers.
Here and there I found some typos and weird wording, but this is just a minor concern.
Author Response
The authors are grateful to the reviewer for his/her comments that are individually answered below:
- My feeling is that the development is mainly focused on software, which exploits the powerful features of the Red Pitaya board. Indeed the authors have done a relevant job, but I have a few perplexities.
- Apparently there is no development of sensors, just a complex software development that by the way looks prominent and makes use of other existing sensor(s) and the 125 MHz sampling ADC onboard of Red Pitaya.
The submitted paper may have given this impression, but, even if the firmware and the software development are an important part in the described activities, the reported work addresses also the integration of the measurements in a large network of sensors . In order to better clarify this in the revised paper, (1) the reasons of the proposed approach have been expanded (lines 95-104), (2) a section has been added, that describes how the new boards have been integrated in the central data acquisition system(lines 321-338) and (3) the application section has been expanded to provide a more complete description of one of the applications of the sensors whose outputs are acquired by the RP boards (lines 370-417)
- The description of the work done focuses on the developed software (including FPGA programming), with some interesting features like the automatic driver generation.
- However, such a description is too specialistic and makes use of too many acronyms and jargon, not always explained.
The paper has been changed in several parts to simplify somewhat the provided descriptions and explaining acronyms that previously were given as assumed. This should improve readability also for non-experts in the field.
line 45
...sampling rate FOR such transient phenomena... Fixed
lines 173-189
This paragraph sounds unclear, I think it needs to be rewritten in a more schematic fashion, better specifying the roles and locations of Pre and Post pointers, of the samples, who and how moves them.
e.g. Two pointers, PrePtr and PostPtr are used: the second one is moved to the next position... Does "is moved" mean physically copied elsewhere or simply that its value is incremented in order to point elsewhere?
Moreover, it is not clear whether the samples are on the circular buffer or elsewhere and the pointers are used to locate them.
That section has been rephrased to better describe the algorithm and how the pointers are used
lines 204-223
Too many acronyms, the general reader will be immediately lost. In order to reach a larger number of readers the content should be more understandable. The way it is now it is basically reserved to the real experts of low-level software developers.
An effort to simplify the text, or in alternative to improve the description, would be beneficial.
Table 1
Same as above for the table.
As before, the text has been simplified, better explaining acronyms that were before assumed as already known by the reader.
Lines 233-238
Again, this sounds like low-level programmers jargon. For instance I doubt that the general reader will understand the statement "...due to limitation of available LUTs components in the Zynq 7010 SoC used in the RP board."
I do not know the Red Pitaya system and even though I have a solid experience in low-level programming I struggle to follow the text that seems directed to experts of Red Pitaya.
The sentence has been rephrased in order to make it more readable
Lines 306-311 and fig.4
Closed Loop Hall Effect transducers?
rogowsky coils?
What are these transducers? Any reference?
Moreover, the figure as it is is just for the show: very small, no details, totally incomprehensible.
Lines 327-332
Sounds like nerd jargon, not for general readers.
The adopted names were unfortunate and may have mislead the reader. Now, the correct naming convention (Wideband Current Transformers and Closed Loop Direct Current Current Transformers ) has been used (giving also two references), addressing well known technologies for measuring alternate and continuous currents in conductors. The purpose of the figure is not to explain how the sensor works, but how it is connected to the data acquisition board.
Reviewer 3 Report
Comments and Suggestions for Authors
The paper presents a solution for high-speed acquisition systems design based on the OEM components, while pointing at the parts where the system can be customized. I think the article might be interesting to the scientists that needs to build acqusisition systems for their research but would like to limit the time spent for engneering work and system development. Although similar approach has been already presented in other articles in the field, this paper can help scientists find another, alternative tools to do so.
I have following comments:
1. Line 34 - please, explain what CODAS is. Explain the abbreviation or give a reference.
2. Line 106 - 109 - I'd suggest the Authors to look for some references to other articles that present alternative solutions to the one in this paper, meaning the solutions that enables quick development of high-speed acquisition systems for nuclear science based on the off the shell devices and platforms/libraries. It can be on Zynq, too, but there are alternatives, e.g. with the use of National Instruments cards and environment. Please, add few references to the text.
3. Line 201 - I believe you lost "b)"
4. Line 220 - I believe you meant FIFO instead of FIO. If not, please explain what FIO is.
5. Line 255 - You lost dot after word "processor"
6. Figure 2 - Can you label Y axis? Additionally, I'd suggest logarithmic Y axis. Also, in line 267 you refer to the maximum sampling rate of 125 MHz while X axis is from 1.2 MHz to 2.2 MHz
Comments on the Quality of English Language
7. Line 258 - probably "DMA-based" would be better
8. Line 265 - "In the figure," - comma after figure
Author Response
The authors are grateful to the reviewer for his/her comments that are individually answered below:
Line 34 - please, explain what CODAS is. Explain the abbreviation or give a reference.
The acronym is now explained at line 29
- Line 106 - 109 - I'd suggest the Authors to look for some references to other articles that present alternative solutions to the one in this paper, meaning the solutions that enables quick development of high-speed acquisition systems for nuclear science based on the off the shell devices and platforms/libraries. It can be on Zynq, too, but there are alternatives, e.g. with the use of National Instruments cards and environment. Please, add few references to the text.

In the revised paper it is now stated that, although many commercial ADC solutions exist for streamed data acquisition, only very few commercial solutions are available (to our knowledge) for the specific application (event driven streamed acquisition). A reference has been provided to the commercial solution that most closely reflects out requirements.
- Line 201 - I believe you lost "b)"
 - Line 220 - I believe you meant FIFO instead of FIO. If not, please explain what FIO is.
 - Line 255 - You lost dot after word "processor"

Fixed
- Figure 2 - Can you label Y axis? Additionally, I'd suggest logarithmic Y axis. Also, in line 267 you refer to the maximum sampling rate of 125 MHz while X axis is from 1.2 MHz to 2.2 MHz

Y axis has now been labeled, and in the figure caption it is explained why the frequency range is considered in the figure.
- Line 258 - probably "DMA-based" would be better
 - Line 265 - "In the figure," - comma after figure

Fixed
Reviewer 4 Report
Comments and Suggestions for Authors
This article addresses the high speed data acquisition field with application in nuclear fusion experiments.
Although the article is more technical, the presented solution presents a high level of complexity and serves as a crucial component within the ITER neutral beam test facility.
After reading and analyzing the article, I can make the following observations:
- The current use of the system is for data acquisition. However as authors stated the proposed system can be used also for control. In this case, it is indicated that direct FIFO readout is required (read data samples from FIFO as soon as they are available). Does the circular buffer will still be used in this case? Is it possible that the presence of this buffer to add delays even if the output FIFO is read directly. Maybe both circular buffer and output FIFO must be bypassed for control applications (with reduction of the sampling rate according to FPGA-microcontroller transfer speed limitations).
- In Breakdown measurement application where trigger signal is distributed to all acquisition devices, does the trigger signal is transmitted by WI-FI? If yes, is WI-FI communication able to assure a deterministic propagation time of the trigger to all devices?
- There are some minor spelling issues (at line 182 "is the two pointers" instead "of if the two pointers", line 211 "deriver", line 220 "VIADO")
In conclusion, I recommend to accept the article after minor revisions related to previous observations.
Author Response
The authors are grateful to the reviewer for his/her comments that are individually answered below:
- The current use of the system is for data acquisition. However as authors stated the proposed system can be used also for control. In this case, it is indicated that direct FIFO readout is required (read data samples from FIFO as soon as they are available). Does the circular buffer will still be used in this case? Is it possible that the presence of this buffer to add delays even if the output FIFO is read directly. Maybe both circular buffer and output FIFO must be bypassed for control applications (with reduction of the sampling rate according to FPGA-microcontroller transfer speed limitations).
A further explanation has been added (lines 314-317) clarifying that in order to carry out real-time control behavior, it is necessary to make sure that no internal buffering occurs, and that this can be achieved at sampling frequencies lower than 1.2MHz
- In Breakdown measurement application where trigger signal is distributed to all acquisition devices, does the trigger signal is transmitted by WI-FI? If yes, is WI-FI communication able to assure a deterministic propagation time of the trigger to all devices?
In the conclusions it has been now clearly stated that WiFi has been used to transfer acquired data, but not to exchange triggers. Fiber optics are used instead for triggers.
- There are some minor spelling issues (at line 182 "is the two pointers" instead "of if the two pointers", line 211 "deriver", line 220 "VIADO")
Fixed
Round 2
Reviewer 2 Report
Comments and Suggestions for Authors
Manuscript: 2668287
Title
A versatile board for event driven data acquisition
The manuscript quality was enormously improved, in my opinion.
I confess I doubted the authors would make it, I thought they would accept my other suggestion to submit it to a different, more software oriented, journal.
Nonetheless, they have done a remarkable job, incorporating several parts directly related to sensors and measurements.
I want to hope that my comments helped in producing a higher quality article.